# How Income Inequality and Race Concentrate Depression in Low-Income Women in the US; 2005–2016

**DOI:** 10.3390/healthcare10081424

**Published:** 2022-07-29

**Authors:** Hossein Zare, Adriele Fugal, Mojgan Azadi, Darrell J. Gaskin

**Affiliations:** 1Department of Health Policy and Management, Johns Hopkins Bloomberg School of Public Health, School of Business, University of Maryland Global Campus (UMGC), 624 N. Broadway, Hampton House 337, Baltimore, MD 21205, USA; 2Utah County Health Department, The University of Utah, Provo, UT 84601, USA; adrielef@utahcounty.gov; 3School of Business, University of Maryland Global Campus (UMGC), Largo, MD 20774, USA; mojgan.azadi@faculty.umgc.edu; 4Hopkins Center for Health Disparities Solutions, Department of Health Policy and Management, Johns Hopkins Bloomberg School of Public Health, Baltimore, MD 21205, USA; dgaskin1@jhu.edu

**Keywords:** poverty-to-income-ratio, women, depression, depressive symptoms, racial disparities, income, income inequality

## Abstract

*Aim*: To estimate the association between income and depressive symptoms in adult women, ages 20 years and older. *Methods*: Data for this study came from the 2005–2016 National Health and Nutrition Examination Survey (NHANES). We measured the presence of depressive symptoms by using a 9-item PHQ (Public Health Questionnaire, PHQ-9) and the Poverty to Income Ratio (PIR) as a proxy for income. We employed Negative Binomial Regression (NBRG) and logistic regression models in a sample of 11,420 women. We adjusted models by age, racial/ethnic groups, marital status, education, health insurance, comorbidity, and utilization of mental health professionals. We calculated the Gini Coefficient (GC) as a measure of income inequality, using PIR. *Results*: Between 2005 and 2016, 20.1% of low-PIR women suffered from depression (PHQ ≥10) compared with 12.0% of women in medium-PIR and 5.0% in high-PIR. The highest probabilities of being depressed were in Black Non-Hispanics (BNH) and Hispanics (12.0%), and then in White NH (WNH; 9.1%). The results of NBRG have shown that women in medium-PIR (0.90 [CI: 0.84–0.97]) and high-PIR 0.76 (CI: 0.70–0.82) had a lower incidence-rate ratio than women in low-PIR. The logistic regression results showed that income is protective in High-PIR groups (OR = 0.56, CI [0.43–0.73]). *Conclusion*: Policies to treat depression should prioritize the needs of low-income women of all racial groups and women.

## 1. Background

Depression is a mental health disorder with profound implications for the health and well-being of both individuals and their communities [1]. Depression presents a considerable economic burden in the U.S, costing $US 326 billion in 2018, an increase of 37.9% since 2010 [2] which is slightly higher than GDP growth in the same period (36.9%) [3]. In 2019, an estimated 19 million adults in the U.S. experienced at least one major depressive episode, in [4] and considerable increases in the rate of major depressive disorders (MDD) and co-occurring suicidal ideation (SI) have been observed over the past nine years [5]. Depression has also been associated with increased risk for the development of diabetes mellitus, cardiovascular diseases, hypertension, and respiratory diseases [6]. Women, who are twice as likely as men to develop depression, in [4] experience a multitude of risk factors that increase the likelihood of depression, including hormonal fluctuations during puberty [7] and pregnancy as well as societal factors such as gendered discrimination, socioeconomic inequality, and caretaking responsibilities [8]. However, there remains a critical need to further investigate the impact of racial discrimination and income inequality on depression among women.

While social inequities that lead to economic instability and poverty are common among all genders, the gendered difference in depression rates extends even among those living below the poverty line, with 19.8% of women experiencing depression compared to 15.8% among men [9]. Further, women experience higher rates of socio-economic difficulties, as they are 53% more likely than men to live in poverty [10], and earn 84% of what men earn [11]. Higher depression rates are seen among homeless women, low-income mothers with young children, and women who face food insecurity [12]. Depression among women may be linked to the Cumulative Disadvantage theory, which posits that individuals who have had an early advantage in life will accrue advantage as they age, while individuals who have lived at a disadvantage early in life will experience worse health outcomes, including depression, as they age [12]. Women are also disproportionately more likely to be primary caregivers (66% of women), in [13] which has been linked to lost income and time, and lack of health insurance [14]. Access to insurance can also play a pivotal role in treatment, as all adults with mental illnesses who were enrolled in health plans through employers spent an average of $1347 for out-of-pocket services, while adults without mental health illnesses spent $671 [15]. Such economic difficulties have also been linked to lack of treatment, as 35% of adults with a serious mental health illness reported that they had not received mental health care in the past year [15].

These socioeconomic factors are further complicated by considerations of race and racial discrimination. While several studies have found significantly higher rates of depression among non-white older adults, in [16] there remains a critical need to further examine racial discrimination as a contributing factor to depressive symptoms, diagnosis, and treatment [17]. A few studies have examined the rates of pregnancy-related depression among diverse cohorts and found that racial discrimination and income inequality were key contributors to higher rates of depression among non-white women during pregnancy [18]. In particular, there is a paucity of research focusing on women in individual ethnic groups, despite the evidence that non-white women, including Native American, African American, and Hispanic/Latino women, experience higher levels of depression and other comorbidities than their white counterparts [19]. Further, there has been little differentiation between individual national backgrounds and cultures within larger race and ethnicity groups despite findings that the rates of depression may vary substantially within racial subgroups (e.g., Chinese-Americans compared to Korean Americans) [20]. The situation becomes even more dismal when comparing health care access among minorities, as they are more likely to delay or not seek treatment for their mental illnesses, or if they do, they are less likely to receive appropriate treatment compared to White Non-Hispanic [21]. Also, racial and ethnic minorities are more likely to see care for primary care physicians and other community-based mental health providers compared to psychiatrists and psychologists [21].

## 2. Aims

Despite a fair amount of research regarding the impact of income and health, in [22,23] there is a gap in the literature regarding the relationship between income inequality and depression. This study seeks to examine the association between income and depressive symptoms among women and how this association may operate differently within specific race and ethnicity groups; we will also compare the income inequality and concentration of depression among race and ethnicity groups. This is the first study that uses longitude data to show depression among women of color, to the best of our knowledge.

## 3. Methods

### 3.1. Data and Study Population

We used the 2005–2016 National Health and Nutrition Examination Survey (NHANES) for this analysis. The NHANES is a cross-sectional survey that represents national estimates of the health and nutritional status of the US population; its response rate was 73.2% between 1999 and 2016 [24,25]. This dataset benefits from a multistage probability sampling design to make it representative of the four US regions. For this study, we included a population who were 20 years old and older, and who identified themselves as White Non-Hispanics (WNH), Black NH (BNH), and Hispanics. The NHANES has not reported some control variables for the younger population, so we chose to use the 20 years and older population to have a higher sample size. We excluded missing observations for variables that had been used in the regression models (1478 women), which yielded an analytic sample of 11,420 including 5666 White Non-Hispanics (WNH), 2617 Black Non-Hispanics (BNH), and 3137 Hispanics.

### 3.2. Measures

Dependent variable: The NHANES used the nine-item Public Health Questionnaire (PHQ-9) to measure depressive symptoms. It is a well-known clinically validated survey with a sensitivity and specificity of 88% [26]. PHQ-9 is considered to have a predictive value superior to other screening tools.

The PHQ-9 has measured depressive symptoms of participants in the past two weeks, including participants experiencing restless sleep, poor appetite, and feeling lonely. Each item was scored on a 4-point ordinal scale for frequency (0, not at all; 1, several days; 2, more than half the days; 3, nearly every day). This approach yielded a maximum score of 27 as the sum of the 9 items [27]. For this study, we used this scale as the dependent variable in all Negative Binomial Regression (NBRG) models. Additionally, to run a logistic model, we created a dummy variable with a cutoff score of 10 or higher (=1, if PHQ≥10; =0 if PHQ <10), studies have used this cut-point to measure the clinical depression [26].

### 3.3. Main Independent Variable

Poverty Income Ratio (PIR): The main independent variable of interest was income; we used the PIR as a proxy for income. The PIR was calculated by dividing family (or individual) income to the poverty guidelines specific for each survey year. For example, for a family of four with 5 children younger than 18 years of age with an annual income of $33,935 and a poverty threshold of $35,499, the PIR would be ($33,935/$35,499 = 0.956) [28]. Originally this measure is a continuous variable, we created a categorical variable with 3 categories: low-PIR (0–1.16), medium-PIR (1.17–2.82), and High-PIR (2.83–5.00) quintiles from low to high.

### 3.4. Covariate

We controlled models for demographic variables such as age (years), marital status (1, if married; 0, otherwise), race and ethnicity (WNH, BNH, and Hispanics), educational attainment (less than high school graduate, high school graduate or general equivalency diploma, more than high school education or some college and above). Additionally, all models were controlled for comorbidity (any type of chronic disease), health insurance coverage (1, if had any type of health insurance coverage; 0, otherwise), visiting a mental health professional (1, if had any visit during last 6 months; 0, otherwise), and if a participant had any place to received routine health care (1, if yes; 0, otherwise).

### 3.5. Analytic Strategy

We used descriptive analysis to compare the mean and proportional differences between race and ethnicity groups for all study variables. Demographics, SES, and depressive symptoms were evaluated using ANOVA/chi-square.

To find the best fit model, first, we ran a Poisson regression model; we learned that the Poisson model is inappropriate because the value of goodness of fit chi-square was (48,187, *p* < 0.01) [29]. We then decided to conduct several sets of weighted negative binomial regression models, in [30] to estimate the impact of the PIR on participants’ PHQ-9 scores. The likelihood ratio test showed that NBRG was more appropriate than the Poisson regression model to address the overdispersion (chi-sq = 20,000, *p* < 0.001). The NBRG reports Incidence-Rate Ratios (IRR) and the corresponding 95% confidence intervals (CI) [31,32].

For the first model (basic model), we just included PIR, age and race. For the second model we added marital status, education, speaking English professionally. For the third model we added comorbidity, having health insurance, seeing mental health professionals, a routine place to go for health care, and being employed. We interacted PIR categories and race/ethnicity and because the interaction was significant (*p* < 0.001), we stratified the full models by race/ethnicity. 

Additionally, to estimate the probability of being depressed and the impact of the PIR, we created a dummy variable (by using participants’ PHQ-9 scores) with a cutoff score of 10 or higher (=1, if PHQ ≥ 10; =0, if PHQ < 10) and ran sets of logistic regression models.

Finally, we used the PIR to estimate the Gini coefficient (GC) as a well-known measure of income inequality. The GC is defined as A/(A + B): A is defined as the area between the line of perfect equality (45-degree line) and the Lorenz Curve, and B is the area between the Lorenz Curve *x*- and *y*-axis, with an ‘A’ equal zero, we will have a zero GC, which stands for perfect equality and with a zero value for the ‘B’ the GC will be one, which explains absolute inequality [33]. We presented the PIR distribution among the depressed and non-depressed using the Lorenz curve. Also, we added a concentration curve to show the prevalence of depression among PIR in WNH, BNH, and Hispanics. The concentration curve plots the cumulative percentage of the clinically depressed population (PHQ ≥ 10, *y*-axis) against the cumulative percentage of the people. We ranked by PIR, beginning with the lowest PIR, ending with the richest (*x*-axis), and the 45-degree line shows perfect equality [34].

All analyses were weighted using the NHANES individual-level sampling weights for 2005–2016 (a cross-sectional analysis of 6 waves of pooled data). As such, the estimates were representative of the national level for the US civilian population.

## 4. Results

### 4.1. Descriptive Analysis Results

We included 11,420 individuals in our analyses. Among all participants, 18.0% (*n* = 3156) had low-PIR, 30.6% (*n* = 4025) had medium-PIR, and 51.5 had high-PIR (*n* = 4323). A total of 75.3% of the study population were WNH, 11.9% BNH, and 12.8% Hispanics. The study population’s average age was 49.0 years (SD = 14.5), 60% of participants were married, and 63% had a degree beyond a high school diploma (See Table 1).

Comparing the SES variables among race/ethnicity groups showed that WNHs were older (50.3 ± 11.7), more likely to have a higher PIR (59%), had health insurance (90%), a routine place to go for health care (93%), mental health professional visits (10%), and comorbidity (1.24). BNHs had the lowest rate of being married (38%), the lowest rate of having a college degree or above (20%) after Hispanics (13%), and the highest rate of having comorbidities after WNHs. The Hispanic populations were the youngest (43.0; SD: 19.0) group, with the lowest rate of comorbidity and the lowest rate of having health insurance coverage.

The average score of PHQ-9 was 3.48 (SD: 3.70), with the highest score in BNH (3.90, SD: 5.59) and Hispanics (3.84, SD: 3.70), and the lowest score in WNH (3.36, SD: 2.87).

### 4.2. Depressive Symptoms and PIR Categories

In Figure 1, we presented the distribution of PHQ-9 among race and ethnicity groups. For this figure, we used the average percentage of women who had a PHQ score equal to or larger than 10. As shown in all race and ethnicity groups, the low-PIR population experienced a higher rate of being diagnosed with depression, with the highest probability of being depressed in BNH and Hispanics (12.0%), and then WNH (9.1%).

### 4.3. Association between Poverty Income Ratio Levels and Depressive Symptoms in NBRG

Table 2 presents the results of NBRG models to estimate the association between PIR levels and PHQ-9 scale. In comparison with the low-PIR group, participants in medium- and high-PIR had a lower incidence-rate ratio (IRR). The results of the adjusted full-model indicated that the incident rate for medium-PIR was 0.90 (CI: 0.84–0.97) times the incident rate for the reference group (low-PIR). The incident rate for high-PIR was 0.76 times (CI: 0.70–0.82) the incident rate for the participants in low-PIR while holding the other variables constant. Income worked as protective for depressive symptoms.

Educational attainment, marital status, having health insurance, and being employed were protective against depressive symptoms, but comorbidity and seeing mental health professionals had a positive association with the higher depressive symptoms.

### 4.4. Association between Depressive Symptoms and Race/Ethnicity

The unadjusted incident rate for BNH women was 1.15 (CI: 1.07–1.24) times the rate for WNH women. Similarly, the unadjusted incident rate in Hispanic women was 1.13 (CI: 1.05–1.22). However, after adjusting for PIR, educational attainment and language, this association by race/ethnicity was statistically insignificant with both IRRs less than 1.

High income was protective against depressive symptoms in WNH, BNH, Hispanics, and other racial groups, but the medium income was protective only in WNH. There were some differences between SES and other control variables. For example, higher education and being employed were always protective, marital status was protective in WNH and Hispanics, and it may be because of a lower rate of being married in BNH. Health insurance was only protective in WNH and could be explained by the higher insurance rate in WNH. People with comorbidity have experienced higher IRR (1.19 (CI: 1.17–1.22)), with almost similar IRRs among racial groups (see stratified models by race in Table 2). Also, visiting mental health professionals (IRR: 1.93 (CI: 1.79–2.09)) was positively associated with higher depressive symptoms and among all race/ethnicity groups. The results of interactions between PIR categories and race/ethnicity showed that PIR was not protective in Black NH and Hispanics.

### 4.5. Association between Poverty Income Ratio Levels and Probablity of Being Depressed

Table 3 presents the results of the logistic regression models. For this analysis, the dependent variable is a dummy variable with a cutoff score of 10 or higher (=1, if PHQ ≥10; =0, if PHQ <10) indicating whether a respondent screened for depression. In the base model we observe race and ethnic differences in the likelihood of screening for depression. The odd ratios of BNH women and Hispanic women screening for depression was (OR = 1.34, CI [1.12–1.60]) and (OR = 1.35, CI [1.13–1.61]), respectively. However, this association is no longer significant and practically one after adjusting for PIR. The full model showed that income is protective in High-PIR groups (OR = 0.56, CI [0.43–0.73]).

Similar to NBRG, there were some differences between SES and other control variables. For example, some colleges and above, being married, having health insurance, and being employed are protective against depression. People with comorbidity have experienced higher OR (1.47 (CI: 1.29–1.66)) of being depressed and visiting mental health professionals were positively associated with higher depressive symptoms (OR: 4.91, CI: [3.23–7.25]).

### 4.6. Depressive Symptoms and Income Inequality

To learn how depressive symptoms moved by income inequality, we have computed the Gini Coefficient (GC) for women with PHQ-9 ≥ 10—these populations clinically are considered as people with depression—and women with PHQ-9 < 10 (clinically no-depression). The Gini Coefficient for the clinically depressed women (PHQ > 10) showed that they have higher GC (0.382, SE: 0.008) than non-depressed GC (0.263, SE: 0.003). We found a significant difference between these two groups. We found similar patterns in BNH with GC of 0.368 (SE: 0.004) in the non-depressed, and 0.433 (SE: 0.012) in the depressed populations. We found a very close GC between non-depressed (0.393 (0.004)) and depressed 0.409 (SE: 0.013) Hispanics population.

### 4.7. Concentration of Depressive Symptoms among Poor and Rich Population

To learn how depressive symptoms concentrated among PIR we plotted the concentrated curve of clinically depress population (PHQ ≥ 10, *y*-axis) against the cumulative percentage of the population for WNH, BNH and Hispanics. Also using the Lorenz curves, we plotted the distribution of PIR.

Figure 2A presents the concentration of depression for WNH populations ranked by PIR. As expected, the concentration curve for depressed (blue-solid line) lies above the equal distribution line. That is a higher probability of being depressed in the population with low income. For example, the bottom 28% of households in the income distribution have experienced about 62.3% of all depression in WNH populations. The red line presents Lorenz Curve for income in WNHs. We see a similar pattern in BNHs, e.g., 21% of households in the income distribution have experienced about 61.1% of all depression of BNH (See Figure 2B) and lower concentration curve in Hispanics, whether 22% of households in the income distribution have experience about 60.0% of all depression of Hispanics. (See Figure 2C). As presented by Lorenz Curve, we see the highest disparity in income between BNHs (e.g., 70% of income distributed among 40% of the population) and Hispanics with almost the same distribution (See red line under 45-degree lines in Figure 2A–C). 

## 5. Discussion

We studied the association between depressive symptoms and income measured by PIR in US adult women 20 years of age and over with a sample of 11,420 people. We used NBRG regression models to estimate the incidence rate ratio of the PHQ-scale and its changes by different levels of income, we also used logistic regression models to estimate the likelihood of being depressed (if PHQ scale ≥10). Our findings showed that low-income women suffered more from depression than higher-income women; they were at a greater risk of suffering disproportionately from depression. For example, 20.1% of low-PIR women have PHQ scale ≥10 in comparison with 12.0% of medium-PIR and 5.0% of high-PIR groups. As reported by prior studies, the higher incidence of depression may be because of the social isolation [35] caused by low income that exacerbated the risk of developing depressive symptoms, or by lower access to mental health services either because of living in areas with low access, having socioeconomic disadvantages, in [14] lack of health insurance coverage, or the inability to pay the cost of treatment. Comparing the classes of depression showed that the low-PIR population more likely to have ‘moderate’ and ‘moderate-severe’ depression, it prioritizes need for immediate attention for these population.

Between 2005 and 2016, for all race and ethnicity groups, low-income populations experienced a higher probability of being depressed, with a 32.3% increase from 13.6% in 2005 to 18.0% in 2016 (See Figure 1 and Appendix A Table A1). We observed the highest probability of being depressed in low-income WNHs (22.3%) and then in BNHs (17.3%) and Hispanics (16.6%) between 2005 and 2016. Considering the cost of Major Depressive Disorder (MDD) in the US (10,379 USD) [2], and with increasing the depression after the Pandemic [36], policymakers should think about mental health screenings with prioritizing low-income populations. Even though there is a higher probability of being depressed in low-income WNHs with lower resources [37,38], and high income inequality [39] in communities of color, it is crucial to improve mental health services, including but not limited to screenings among all low-income women.

*Concentration of**depression among Race and Ethnicity, and Income.* The concentration curves showed that a higher probability of being depressed in the population with low income, among all race and ethnicity groups. Communities of color were more likely to have lower incomes and live in neighborhoods with higher socially disadvantaged rates and lower resources [37,38]. The complexity of depression, income inequality [39] and the racial composition of populations expands with adding the gender element. Indeed, this study is unique for including these elements together. Similar to other studies, our findings showed that women of color suffered a higher level of income inequality and a higher concentration of depression than their White counterparts [26,27,28]. This may be because of delaying treatment for their mental illnesses or not receiving appropriate treatment, in [26,27,28,30] or lower access to mental health services, or unaffordability of mental health treatment. Despite a few studies [40] we found that income is protective, policymakers should prioritize reducing income inequality in low-income populations.

*Higher-income inequality in depressed women.* The GC showed that depressed women with a weighted GC of 0.403 (SE: 0.006) experienced higher income inequality than non-depressed with a GC of 0.301 (SE: 0.002). Comparing GC among race and ethnicity groups showed the higher GC for the non-white population, with the highest GE for the BNH (GC: 0.433, SE: 0.012). We need to stress the importance of income inequalities, not only because of their direct impact on depression, as shown in low-PIR populations, but most importantly because of the indirect impact of income inequalities on health outcomes, “rich may write the rules in their favor, and they may work against the public provision of health care” [41]. Combining gender, race, and income inequality creates more complexity; for example, a study has shown that combining income inequalities and racial disparities can increase psychosocial stress [42]. Socioeconomic disadvantage was another important element. Women with higher depression rates have experienced higher rates of socioeconomic disadvantage. Socioeconomic disadvantage has also been connected to a lower probability of receiving treatment for mental health and explains why the low-PIR population in our study experienced a higher probability of severe depression.

*Fundamental inequality.* Depression is a major public health problem—especially for low-income women. Other mental illnesses and physical disorders often accompany it. Women were twice as likely to develop depression than men, 8–9 with a higher impact on low-income populations [43].

Some fundamental inequalities have enhanced the direct and indirect impacts of depression on women, including the structural inequalities such as wage differences, in [44] occupational risk factors, in [45] and the gap in income [46]. Depression alleviation policies should address these structural inequalities by prioritizing low-income women of color who are living in socioeconomically disadvantaged neighborhoods. Specifically, this population needs immediate attention for a social assistance benefit package to slow down the cumulative disadvantages and prevent developing severe depression in these populations. The study reported similar findings on the impact of income and depression in OECD countries, e.g., using the Belgian Health Interview Survey study has reported household income as a protective element on depression [47], Literature reported greater income equality and higher population health standards [48]. Also, a study showed that countries with greater income inequality recorded higher depressive symptoms [49]. There are lessons to learn from countries such as Finland, New Zealand, the Netherlands, and other Nordic countries with well-developed social welfare systems. The US may take advantage of some economic policies to eliminate income inequality, such as progressive taxation to reduce the wealth gap.

Several aspects of the present study deserve comment. The publicly available NHANES data had some limitations regarding the income variable and did not report real income. Because were not able to use a panel data, we were unable to draw causal inferences. We are aware that employing household income as a continuous variable could allow us to estimate the impact of income differences instead of a proxy variable such as the PIR. Finally, we were not able to control models for geographical variables; evidence showed an association between geographical areas and health outcomes.

There are also strengths to this study. For example, based on our knowledge, this is the first study with a wide range of data between 2005 and 2016 to examine the relationship between income and depression in women. Additionally, for this study, we benefited from the weighted analyses that made our findings nationally representative estimates, as mentioned earlier the NHANES data represents national estimates of the health and nutritional status of the US population.

## 6. Conclusions

Between 2005 and 2016, depression increased by 22.2 percentage points in adult women 20 years of age and older, with a higher increase in low-income populations by 24.4 percentage points. The results of NBRG showed a strong association between depression and PIR in US women 20 years of age and older.

Policymakers should consider a combination of local and federal policies to slow down the depression growth, they should address the structural racism and sexism that undoubtedly lead to health inequities to improve and facilitate access to mental health service including screening to identify mental disorder in early stage. Policies should also consider gender and racial/ethnic inequities. Policies to mandate equal compensation for women and racial/ethnic minorities are critical in closing the pay gap by prioritizing low-income, socially disadvantaged women of color. Developing social assistance and some community-level activities for women who are dealing with depression could keep people with depression more involved in community activities and save more taxpayer money.

## Figures and Tables

**Figure 1 healthcare-10-01424-f001:**
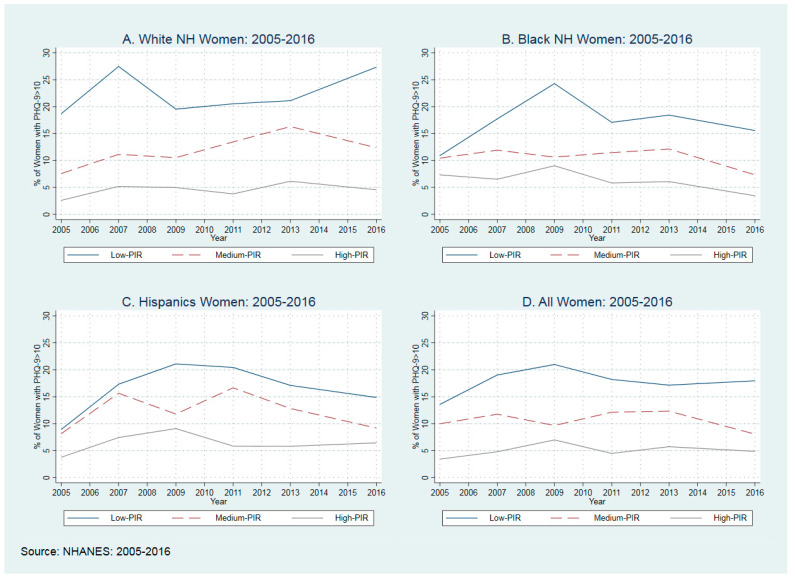
Distribution of Clinically Depressed Women among Race and Ethnicity between 2005–2016.

**Figure 2 healthcare-10-01424-f002:**
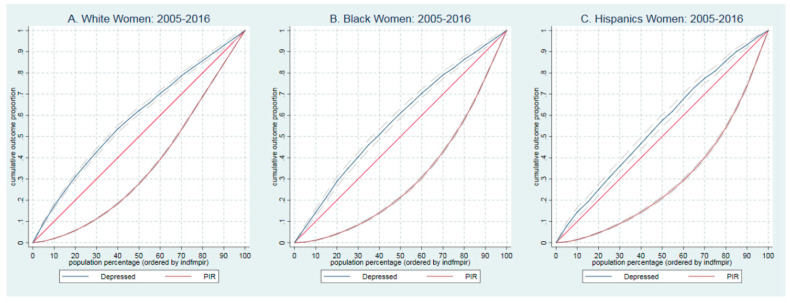
Concentration curves of depression and PIR in women 20 years and older: 2005–2016.

**Table 1 healthcare-10-01424-t001:** Demographic Characteristics of US Women Aged 20 and Above; NHANES 2005–2016.

	White NH (*n* = 5666)		Black NH (*n* = 2617)		Hispanics (*n* = 3137)		All (*N* = 11,420)	
	Mean/%	(SD)	Mean/%	(SD)	Mean/%	(SD)	Mean/%	(SD)
Section A: Depressive Symptoms								
PHQ-D Score	3.36	(2.87)	3.90	(5.59)	3.84	(5.80)	3.48	(3.70)
If PHQ > 10 ^α^	0.09	(0.20)	0.12	(0.38)	0.12	(0.40)	0.10	(0.25)
**PHQ Categories**								
Minimal	0.74	(0.30)	0.70	(0.54)	0.69	(0.57)	0.73	(0.38)
Mild	0.17	(0.25)	0.18	(0.45)	0.19	(0.48)	0.17	(0.32)
Moderate	0.06	(0.16)	0.07	(0.30)	0.07	(0.32)	0.06	(0.20)
Moderate-Severe	0.03	(0.11)	0.03	(0.21)	0.03	(0.22)	0.03	(0.14)
Sever	0.01	(0.06)	0.01	(0.14)	0.01	(0.14)	0.01	(0.08)
**Section B: Income and Socio-demographic**								
**PIR Categories**								
Low PIR (0–1.12)	0.13	(0.23)	0.30	(0.53)	0.36	(0.59)	0.18	(0.33)
Medium PIR (1.13–2.72)	0.29	(0.31)	0.35	(0.56)	0.38	(0.60)	0.30	(0.39)
High PIR (2.73–5.00)	0.59	(0.34)	0.35	(0.55)	0.26	(0.54)	0.52	(0.43)
**Socio-demographic**								
Age (years)	50.33	(11.69)	46.33	(18.72)	42.64	(18.89)	48.58	(14.46)
Married	0.63	(0.33)	0.38	(0.56)	0.61	(0.60)	0.60	(0.42)
**Educational attainment**								
Less than high school	0.11	(0.21)	0.20	(0.47)	0.39	(0.60)	0.15	(0.31)
High school graduate/GED	0.22	(0.28)	0.23	(0.49)	0.20	(0.49)	0.22	(0.35)
Some college or AA degree	0.35	(0.32)	0.37	(0.56)	0.28	(0.55)	0.34	(0.40)
College graduate or above	0.32	(0.32)	0.20	(0.47)	0.13	(0.41)	0.29	(0.39)
**Race/Ethnicity**								
White Non-Hispanic	NA	NA	NA	NA	NA	NA	0.75	(0.39)
Black Non-Hispanic	NA	NA	NA	NA	NA	NA	0.12	(0.27)
Hispanics	NA	NA	NA	NA	NA	NA	0.13	(0.28)
Comorbidity	1.24	(0.99)	1.00	(1.48)	0.77	(1.42)	1.13	(1.19)
Has any health insurance coverage	0.90	(0.21)	0.82	(0.45)	0.63	(0.59)	0.85	(0.30)
If respondent speak English professionally	1.00	(0.01)	1.00	0.00	0.73	(0.55)	0.97	(0.15)
See a mental health professional	0.10	(0.20)	0.09	(0.33)	0.08	(0.34)	0.09	(0.25)
A routine place to go for healthcare	0.93	(0.18)	0.92	(0.31)	0.80	(0.49)	0.91	(0.25)
**Employed**	0.58	(0.34)	0.61	(0.57)	0.58	(0.61)	0.58	(0.42)

Note: Depression symptoms categories calculated using the Patient Health Questionnaire–9: none (0–4), mild (5–9), moderate (10–14), moderately severe (15–19), and severe (20). We combined moderately severe and severe as one group for this table. The percentages are weighted to the population of noninstitutionalized US adults aged 20 years or older. ^α^ We created a dummy variable if the score was equal to or higher than 10. NA: Not Applicable.

**Table 2 healthcare-10-01424-t002:** Weighted Negative Binomial Regression Estimates in US Women Aged 20 and above; NHANES 2005–2016.

	Basic Model	2nd Model	3rd Model	Stratified Models by Race and Ethnicity		
	*n* = 11,420	*n* = 11,420	*n* = 11,420	White NH *n* = 5666	Black NH *n* = 2617	Hispanics *n* = 3137
	IRR [95% CI]	IRR [95% CI]	IRR [95% CI]	IRR [95% CI]	IRR [95% CI]	IRR [95% CI]
**PIR Categories (Ref: if PIR < 1.12)**						
Medium PIR (1.13–2.72)		0.80 ***	0.90 **	0.88 *	0.91	0.94
		[0.74–0.86]	[0.84–0.97]	[0.80–0.97]	[0.81–1.03]	[0.85–1.04]
High PIR (2.73–5.00)	NA	0.59 ***	0.76 ***	0.73 ***	0.83 **	0.82 *
		[0.54–0.63]	[0.70–0.82]	[0.65–0.83]	[0.72–0.95]	[0.70–0.96]
Age	1.00 *	1.00 ***	0.99 ***	0.99 ***	0.98 ***	1.00 *
	[1.00–1.00]	[1.00–1.00]	[0.99–0.99]	[0.99–0.99]	[0.98–0.99]	[0.99–1.00]
Married	NA	0.83 ***	0.86 ***	0.85 ***	0.93	0.87 *
		[0.78–0.88]	[0.81–0.91]	[0.79–0.92]	[0.84–1.03]	[0.78–0.97]
**Education (Ref. Less than high school)**						
High school graduate/GED or equivalent	NA	0.85 ***	0.84 ***	0.84 *	0.71 ***	0.92
		[0.77–0.93]	[0.76–0.93]	[0.74–0.97]	[0.62–0.81]	[0.80–1.05]
Some college or AA degree	NA	0.82 ***	0.79 ***	0.77 ***	0.74 ***	0.9
		[0.75–0.90]	[0.72–0.86]	[0.68–0.88]	[0.64–0.85]	[0.79–1.01]
College graduate or above	NA	0.68 ***	0.68 ***	0.66 ***	0.72 ***	0.75 **
		[0.62–0.74]	[0.61–0.75]	[0.58–0.76]	[0.60–0.87]	[0.62–0.91]
**Race/Ethnicity (Ref. NHW)**						
Black Non-Hispanic	1.15 ***	0.95	1.03	NA	NA	NA
	[1.07–1.24]	[0.89–1.02]	[0.96–1.09]			
Hispanics	1.13 **	0.94	1.01	NA	NA	NA
	[1.05–1.22]	[0.87–1.02]	[0.94–1.10]			
Foreign Language-English (=1, if respondent speak English professionally)	NA	1.16 **	1.09	0.24 ***	1.00	1.03
		[1.05–1.30]	[0.98–1.22]	[0.21–0.28]	[1.00–1.00]	[0.92–1.16]
Comorbidity	NA	NA	1.19 ***	1.19 ***	1.22 ***	1.21 ***
			[1.17–1.22]	[1.16–1.22]	[1.17–1.26]	[1.15–1.26]
Has any health insurance coverage	NA	NA	0.92 *	0.88 *	0.93	0.97
			[0.85–1.00]	[0.79–0.98]	[0.80–1.08]	[0.86–1.09]
See a mental health professional	NA	NA	1.93 ***	1.89 ***	2.06 ***	1.97 ***
			[1.79–2.09]	[1.71–2.09]	[1.80–2.37]	[1.71–2.27]
A routine place to go for healthcare	NA	NA	0.96	0.92	0.90	1.04
			[0.87–1.05]	[0.82–1.05]	[0.74–1.09]	[0.90–1.21]
Employed	NA	NA	0.84 ***	0.85 ***	0.74 ***	0.85 ***
			[0.79–0.89]	[0.79–0.92]	[0.64–0.86]	[0.78–0.93]
Constant	3.69 ***	6.46 ***	7.12 ***	35.58 ***	11.31 ***	4.62 ***
	[3.40–4.00]	[5.53–7.54]	[6.10–8.31]	[29.36–43.12]	[8.18–15.66]	[3.82–5.58]
ln alpha	1.32 ***	1.21 ***	1.05	1.00	1.11 *	1.23 ***
	[1.26–1.39]	[1.14–1.27]	[0.99–1.11]	[0.93–1.07]	[1.01–1.22]	[1.13–1.34]

* *p* < 0.05, ** *p* < 0.01, *** *p* < 0.001. IRR = Incidence-Rate Ratios, NA: Not Applicable.

**Table 3 healthcare-10-01424-t003:** Weighted Logistic Regression to Estimates the Association Between Depression and PIR in US Women Aged 20 and above, NHANES 2005–2016.

	Model 1 (*n* = 11,420)	Model 2 (*n* = 11,420)	Model 3 (*n* = 11,420)	Model 4 (*n* = 11,420)
	Exp(b) [95% CI]	Exp(b) [95% CI]	Exp(b) [95% CI]	Exp(b) [95% CI]
**PIR Categories (Ref: if PIR < 1.12)**				
Medium PIR (1.13–2.72)	NA	0.56 ***	0.65 ***	0.87
		[0.47–0.67]	[0.53–0.79]	[0.68–1.11]
High PIR (2.73–5.00)	NA	0.21 ***	0.32 ***	0.56 ***
		[0.18–0.25]	[0.26–0.40]	[0.43–0.73]
Age	0.99 *	0.99 *	0.99 *	0.98 **
	[0.99–1.00]	[0.99–1.00]	[0.98–1.00]	[0.97–0.99]
Married	NA	NA	0.55 ***	0.55 ***
			[0.43–0.70]	[0.40–0.75]
**Education (Ref. Less than high school)**				
High school graduate/GED or equivalent	NA	NA	0.78	0.74
			[0.58–1.05]	[0.52–1.05]
Some college or AA degree	NA	NA	0.68 **	0.63 **
			[0.52–0.89]	[0.46–0.86]
College graduate or above	NA	NA	0.34 ***	0.36 ***
			[0.24–0.49]	[0.22–0.58]
**Race/Ethnicity (Ref. NHW)**				
Black Non-Hispanic	1.34 **	1.08	0.98	1.15
	[1.12–1.60]	[0.87–1.34]	[0.75–1.28]	[0.85–1.55]
Hispanics	1.35 **	1	1.01	1.26
	[1.13–1.61]	[0.82–1.22]	[0.79–1.29]	[0.91–1.76]
Foreign Language-English	NA	NA	1.2	1.08
			[0.84–1.71]	[0.71–1.62]
Comorbidity	NA	NA	NA	1.47 ***
				[1.29–1.66]
Has any health insurance coverage	NA	NA	NA	0.66 **
				[0.49–0.89]
See a mental health professional	NA	NA	NA	4.91 ***
				[3.32–7.25]
A routine place to go for healthcare	NA	NA	NA	1.13
				[0.78–1.65]
Employed	NA	NA	NA	0.68 *
				[0.49–0.96]

* *p* < 0.05, ** *p* < 0.01, *** *p* < 0.001; Note: NA: not applicable.

## Data Availability

The study does not qualify as human subjects research as defined by DHHS regulations 45 CFR 46.102. and does not require IRB oversight. Available online: https://wwwn.cdc.gov/nchs/nhanes/ (accessed on 9 March 2022).

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
