# Peer review of "How Income Inequality and Race Concentrate Depression in Low-Income Women in the US; 2005–2016"

_healthcare, 2022, doi:10.3390/healthcare10081424_

Round 1

Reviewer 1 Report

I think it is very meaningful as a study of vulnerable population groups.

Thank you for the opportunity to read this well-written study.

Just a few comments.

1. The introduction is well written to reveal the main content. However, in Aims, it is described as racial/ethnic groups and racial/ethical groups. What does it mean?

2. Please provide the number of subjects for all data in the “Data and study population section”.

3. In the discussion, the following sentence needs to be corrected.

 “Comparing the classes of depression showed that the low-PIR population was more likely to have moderate and moderate-severe depression that prioritizes the need for immediate attention.”

As a longitudinal study, it is difficult to estimate a causal relationship.

4. In the discussion, the following sentence needs to be corrected.

 “Despite a few studies with different data set and sample size, 39 we found that income is protective, any effort to reduce the income inequality gap and can reduce the income inequality in favor of the non-White population.”

Somewhat difficult to understand. Also, it is not logically connected to the above paragraphs, so it needs to be corrected.

5. Also, comparisons with men in the discussion are somewhat unreasonable. It is more appropriate to compare and analyze the study with women, the subjects of this study.

6. Also, the following sentence needs to be corrected.

“we could not control geographical variables using the publicly available data, evidence showed that place matters.”

This statement can make the data appear to be non-representative. The representativeness of the data should be emphasized.

Author Response

Reviewer #1

  1. The introduction is well written to reveal the main content. However, in Aims, it is described as racial/ethnic groups and racial/ethical groups. What does it mean?

Response #1:  Thank you for noticing this, we have revised the sentence to address this comment.

  1. Please provide the number of subjects for all data in the “Data and study population section”.

Response #2: Thank you we have added the analytical sample.

  1. In the discussion, the following sentence needs to be corrected.

 “Comparing the classes of depression showed that the low-PIR population was more likely to have moderate and moderate-severe depression that prioritizes the need for immediate attention.”

Response #3:  We have edited this sentence to:

Comparing the classes of depression showed that the low-PIR population more likely to have ‘moderate’ and ‘moderate-severe’ depression, it prioritizes need for immediate attention for these population.

  1. As a longitudinal study, it is difficult to estimate a causal relationship.

Response #4: We added a sentence to the limitation section to address this comment.

  1. In the discussion, the following sentence needs to be corrected.

 “Despite a few studies with different data set and sample size, we found that income is protective, any effort to reduce the income inequality gap and can reduce the income inequality in favor of the non-White population.”

Somewhat difficult to understand. Also, it is not logically connected to the above paragraphs, so it needs to be corrected.

Response #5: We have edited the sentence to:

Despite a few studies40 we found that income is protective, policymakers should prioritizing reducing income inequality in low-income population.

  1. Also, comparisons with men in the discussion are somewhat unreasonable. It is more appropriate to compare and analyze the study with women, the subjects of this study.

Response #6: We agree with the reviewer, we have edited the sentence.

We have copied it here:

Combining gender, race, and income inequality creates more complexity; for example, a study has shown that combining income inequalities and racial disparities can increase psychosocial stress.42 Socioeconomic disadvantage was another important element. Women with higher depression rates have experienced higher rates of socioeconomic disadvantage. Socioeconomic disadvantage has also been connected to a lower probability of receiving treatment for mental health and explains why the low-PIR population in our study experienced a higher probability of severe depression.

  1. Also, the following sentence needs to be corrected. “we could not control geographical variables using the publicly available data, evidence showed that place matters.” This statement can make the data appear to be non-representative. The representativeness of the data should be emphasized.

Response #7: Thank you for the careful review, we have edited the sentence and added another sentence to address the reviewer comment.

We have copied it here:

Several aspects of the present study deserve comment. The publicly available NHANES data had some limitations regarding the income variable and did not report real income. Because were not able to use a panel data, we were unable to draw causal inferences. We are aware that employing household income as a continuous variable could allow us to estimate the impact of income differences instead of a proxy variable such as the PIR. Finally, we were not able to control models for geographical variables, evidence showed an association between geographical areas and health outcomes.

There are also strengths to this study. For example, based on our knowledge, this is the first study with a wide range of data between 2005-2016 to examine the relationship between income and depression in women. Additionally, for this study, we benefited from the weighted analyses that made our findings nationally representative estimates, as mentioned earlier the NHANES data represents national estimates of the health and nutritional status of the US population.

Dear Reviewers, we thank you once again for your most valuable comments and appreciate having had this wonderful opportunity to learn from you. We hope that our responses have addressed your comments effectively.

Reviewer 2 Report

The article "How income inequality and race concentrate depression in low-income women in the US; 2005-2016" covers a very interesting topic: the association between income inequality and race affect women's depression. The study uses data from the 2005-2016 National Health and Nutrition Examination Survey (NHANES) to examine the association between income and depressive symptoms in a in a sample of 11,420 adult women, looking also at different ethnicities. Several methods are used: Negative Binomial Regression (NBRG) and logistic regression models. All models confirm the results that low-income women present a higher probability of being depressed. 

The literature review, objectives and methods are well addressed. Findings and discussion are appropriate.  Since data are longitudinal, if possible, incorporate some discussion on the evolution from 2005 to 2019. An also some more discussion on different results by race.

Minor spelling errors and some differences in the size of the font letter of the text must be addressed (eg. in pages 1,2,3,9,10,11). Also, note that in the second paragraph of the discussion section, some sentences are repeated and reference numbers do not appear as superindices.

Author Response

Reviewer #2

The article "How income inequality and race concentrate depression in low-income women in the US; 2005-2016" covers a very interesting topic: the association between income inequality and race affect women's depression. The study uses data from the 2005-2016 National Health and Nutrition Examination Survey (NHANES) to examine the association between income and depressive symptoms in a in a sample of 11,420 adult women, looking also at different ethnicities. Several methods are used: Negative Binomial Regression (NBRG) and logistic regression models. All models confirm the results that low-income women present a higher probability of being depressed. 

The literature review, objectives and methods are well addressed. Findings and discussion are appropriate.  Since data are longitudinal, if possible, incorporate some discussion on the evolution from 2005 to 2019. An also some more discussion on different results by race.

Response #1:  Thank you for the valuable comments, we added a paragraph in discussion section and an appendix to address this comment.

We have copied it here:

Between 2005-2016, for all race and ethnicity groups, low-income populations experienced a higher probability of being depressed, with a 32.3% increase from 13.6% in 2005 to 18.0% in 2016 (See Figure 1 and Appendix 1). We observed the highest probability of being depressed in low-income WNHs (22.3%) and then in BNHs (17.3%) and Hispanics (16.6%). Considering the cost of Major Depressive Disorder (MDD) in the US (10,379 USD)2, and with increasing the depression after the Pandemic36, policymakers should think about mental health screenings with prioritizing low-income populations. Even though there is a higher probability of being depressed in low-income WNHs with lower resources37,38, and high income inequality39 in communities of color, it is crucial to improve mental health services, including but not limited to screenings among all low-income women.

Minor spelling errors and some differences in the size of the font letter of the text must be addressed (e.g., in pages 1,2,3,9,10,11). Also, note that in the second paragraph of the discussion section, some sentences are repeated, and reference numbers do not appear as superinduces.

Response #2:   We thank you the reviewer for the careful review, we have edited the manuscript to address this comment.

Dear Reviewers, we thank you once again for your most valuable comments and appreciate having had this wonderful opportunity to learn from you. We hope that our responses have addressed your comments effectively.
